# Hybrid Spreadable Cheese Analogues with Faba Bean and Mealworm (*Tenebrio molitor*) Flours: Optimisation Using Desirability-Based Mixture Design

**DOI:** 10.3390/foods12071522

**Published:** 2023-04-04

**Authors:** Laura Garcia-Fontanals, Raquel Llorente, Juanjo Valderrama, Sergio Bravo, Clara Talens

**Affiliations:** 1AZTI, Food Research, Basque Research and Technology Alliance (BRTA), Parque Tecnológico de Bizkaia, Astondo Bidea, Edificio 609, 48160 Derio, Bizkaia, Spain; 2Basque Culinary Centre, Facultad de Ciencias Gastronómicas, Mondragon University, 20009 Donostia-San Sebastián, Spain; 3Blendhub Innovation Department, San Ginés, 30169 Murcia, Spain

**Keywords:** hybrid cheese, faba bean protein, insect protein, desirability-based mixture design, spreadability texture analysis, sensory analysis

## Abstract

Hybrid products could help bridge the gap as new alternative diets emerge in response to the demand for less animal protein, while recent studies suggest that the Western population is not yet ready to fully embrace an alternative protein-based diet. This study used a desirability-based mixture design to model hybrid spreadable cheese analogues (SCAs). The design combined milk protein concentrate (MPC), *Tenebrio molitor* (IF) and faba bean (FBP) flours, representing 7.1% of the formula. Nine SCAs with different MPC/FBP/IF ratios were formulated. Incorporating the IF negatively impacted the desirable texture properties. The FBP flour improved the texture (increasing firmness and stickiness and decreasing spreadability), but only when combined with MPC. Sensory analysis showed that hybrid SCAs (≤50% MPC) C2, C7 and C9 had a more characteristic cheesy flavour than the commercial plant-based reference, and sample C2 had a texture profile similar to the dairy reference. Samples containing IF (C7 and C9) showed a better flavour profile than that without IF (C2). The SCAs had higher protein and lower saturated fat, starch and sugar content than commercial analogues. The study suggests that incorporating alternative proteins in hybrid products can be an effective approach to reduce animal protein content, specifically dairy, in food formulations.

## 1. Introduction

The severity of overpopulation and climate change has motivated the search for new sources of protein to combat food insecurity [1]. In Western countries, food preferences are changing because of the increasing concerns about the environment and health and the growing demand for a reduction in animal protein consumption [2,3]. Alternative diets are emerging to meet this demand. However, the Western population is not ready to drastically change its diet or replace it with plant-based or alternative-protein-based foods. Even so, the need for affordable, appealing alternative products with familiar tastes is often voiced [3].

Hybrid products, partially replacing animal protein with alternatives, might meet this dual demand: to reduce the environmental footprint of food manufacture and offer affordable, familiar, healthy and tasty products [3,4]. Hybrid foods containing dairy or egg ingredients might emerge as a new offer during the transition to an alternative-protein-based diet; using these animal protein sources contributes less to climate change than meat products [3]. 

The market for plant-based dairy alternatives is growing fast [5]. However, the reformulation of conventional dairy products to substitute animal fats and proteins remains challenging. Among dairy products, cheese is a widely consumed food. It has a high content of protein, saturated fats and calories [2], which makes it a good candidate for reformulation. Consequently, the market for cheese analogues is booming. These substitutes could have nutritional and economic advantages over conventional cheese. However, most commercial plant-based cheese analogues on the European market consist of a mixture of water, oils rich in saturated fats, starch and stabilisers. Accordingly, they are very low in protein [6,7,8]. Consumers are aware of the poor nutritional quality of these starch-based analogues and generally believe that these alternatives fail to match the taste and texture of conventional cheese [8].

To develop a hybrid cheese analogue similar to the conventional product, it is necessary to partially replace dairy protein with an alternative protein of similar nutritional quality and functionality. Pulses and insects seem to be interesting protein sources due to their high nutritional quality, techno-functional potential and reduced environmental impact [9]. 

Faba bean is a pulse of great interest as it is a rich source of protein (25–40%), carbohydrate (47–68%) and fibre (11–30%), and it might afford better environmental performance than other legumes. Its cultivation is considered an important method for reducing the environmental footprint; it fixes biological nitrogen, increases the subsequent crop yields, and is easy to cultivate in crop rotation. Faba bean is one of four crops identified by the EU Smart Protein Project as a promising alternative to animal protein. Several studies have evaluated the substitution of animal protein with faba bean protein in food formulations (e.g., bread, pasta and meat and dairy analogues) [9]. Despite its strong nutritional, techno-functional and environmental potential, this plant protein is industrially underexploited as an ingredient for human consumption [10]. 

In contrast, insect proteins have recently received more attention in the food industry due to their high protein content (40–70% on a dry basis), mineral and vitamin contents, and advantageous polyunsaturated to saturated fatty acid ratios. Insect farming also has a reduced environmental footprint compared to livestock. Consumer acceptance is one of the biggest challenges for insect-based food since Western societies are often repulsed by the idea of consuming insects. However, when insect body parts are not evident, the idea seems to create less aversion. This suggests that incorporating the insect in a non-apparent form, such as flour, might facilitate insect-based products acceptance [11,12]. Moreover, Michel and Begho [13] have noticed that consumers preferred insect flour when it was mixed with pork protein, demonstrating that it might be easier to introduce consumers to insect-based foods through hybrid products. 

*Tenebrio molitor* (*T. molitor*) larvae are now more likely to be included in the Western diet after the EFSA declared them safe for human consumption [11]. Research on the techno-functional properties of mealworm proteins and their inclusion in food products has been quite extensive in recent years (e.g., meat products, snacks and bakery) [11,12]. 

To the best of our knowledge, only one study has examined the introduction of insect protein in dairy substitutes [5]. Another group has evaluated the use of faba bean flour in making plant-based cheese analogues [8]. Data on utilising faba bean protein, insect protein or their combination to produce plant-based spreadable cheese analogues or to develop other dairy alternatives are lacking.

The mixture design methodology is used to determine the optimal relative proportion of ingredients using effective information from a relatively small number of experiments. Earlier studies by Talens, Llorente [14] and Talens, Lago [15] have applied this approach to protein blend design for modelling the food product and ingredients. The desirability function approach serves to optimize the balance between the multiple desirable properties of a food product. It is more efficient than traditional methods also used for simultaneous optimisation of several responses; fewer formulations have to be evaluated, and it results in more desirable property levels [16].

The main aim of this study was to use desirability-based mixture design approach to develop a hybrid spreadable cheese with nutritional value, texture and flavour similar to those of conventional dairy spreadable cheese. The study also evaluates the changes in the textural, sensory and nutritional properties of a spreadable cheese analogue (SCA) with dairy protein source replaced by the insect (*T. molitor*) and faba bean flours. The desirability function is used to select the protein ingredient combinations that optimise the prediction model based on the desirable properties of the product.

## 2. Materials and Methods

### 2.1. Raw Materials

Vegetable margarine (70% fat) and salt were purchased at a Makro store (Erandio, Spain). Milk protein concentrate (MPC) (70%) and a mixture of stabilisers were provided by Blendhub S.L. (Murcia, Spain). The MPC was composed by dairy protein and powdered milk and it contained 70% protein, 1.75% fat (1.25% saturated fatty acids (SFA) and 0.5% unsaturated fatty acids (UFA)), and <6% moisture. The stabiliser mix was based on carboxymethylcellulose (E-466), carrageenan (E-407), locust bean gum (E-410), sugar and maltodextrin, and it contained <12% of moisture. Insect flour was obtained from Insekt Label Biotech, S.L. (Bilbao, Spain). The mealworm (*Tenebrio molitor*) flour (IF) composition was 52.2% protein, 33.0% fat (8.0% SFA and 25.0% UFA), 3.7% fibre, 0.4% salt and 10.7% moisture. Faba bean protein concentrate (FBP) was obtained from AGT Foods (Regina, SK, Canada). It contained 60% protein, 3.2% fat (0.6% SFA and 2.6% UFA), 14.4% dietary fibre, 5.3% starch and 10% moisture. Citric acid and sodium citrate were obtained from APASA (Renteria, Spain). Cylindrical glass jars of 128 mL and white lids were acquired from Berlin Packaging Juvasa (Sevilla, Spain).

### 2.2. Pre-Treatment of Insect Flour

As the large particles in the insect flour, reported in previous studies by Talens, Lago [15] and Talens, Llorente [14], can affect the technical properties during hydration and the texture and appearance of the spreadable cheese, the mealworm powder was milled using an ultracentrifuge mill (ZM 100, Retsch, Haan, Germany) to reduce the particle size to ≤200 µm. 

### 2.3. Physicochemical Analysis of Protein Ingredients

The particle size distribution of MPC, FBP and IF was obtained employing a Static Light Scattering Instrument Master-Sizer 3000 (Malvern Instruments Ltd., Malvern, UK) using bi-distilled water as a dispersion agent (refractive index = 1.33), following other studies by Talens, Lago [15] and Talens, Llorente [14]. The d50 (µm) or median particle size by volume (maximum particle diameter of 50% of the sample volume) was obtained. The most common percentiles found were the d10 (µm), d50 (µm) and d90 (µm). The volume moment mean or De Brouckere mean diameter (D [3,4]) was also obtained as it reflected the size of the particles that constitute the bulk of the sample volume. This parameter is most sensitive to large particulates in the size distribution.

The swelling capacity of the flours was determined following Talens, Lago [15] and Talens, Llorente [14]. Water solubility index (WSI), water-holding capacity (WHC) and oil-holding capacity (OHC) were calculated using Equations (1)–(3), respectively:WSI (%) = (Weight of dissolved solid in supernatant)/(weight of dry solids) × 100(1)
WHC (g/g) = (Wet sample weight − dry sample weight)/(dry sample weight)(2)
OHC (cm^3^/g) = (Volume of free oil − initial volume of oil)/(wet sample weight)(3)

The colour of the ingredients was measured using a colourimeter (CR-400 Chroma Meter, Konica Minolta Inc., Tokyo, Japan) in the CIE L* a* b* system. The colourimetric parameters L* (lightness), a* (redness/greenness) and b* (yellowness/blueness) were determined.

### 2.4. Spreadable Cheese Analogues

#### 2.4.1. Experimental Design and Spreadable Cheese Analogues Preparation

A three-factor simplex lattice design with three variables was used to formulate all the possible mixtures (9 assays) containing milk protein concentrate, faba bean protein concentrate and mealworm flour, adding up to 7.1% of the total weight of the ingredients. Within this 7.1%, the maximum content of FBP and IF was 50%, and of MPC was 100%. For the three ingredients, the minimum representation possible was 0% (Figure 1).

The nine different SCA formulations, labelled from C1 to C9 (Figure 2), contained different percentages of MPC, FBP and IF: C1 (7.1, 0, 0), C2 (3.55, 3.55, 0), C3 (0, 3.55, 3.55), C4 (3.55, 0, 3.55), C5 (5.33, 0, 1.77), C6 (5.33, 1.77, 0), C7 (1.77, 3.55, 1.77), C8 (1.77, 1.77, 3.55) and C9 (3.55, 1.77, 1.77). However, for convenience, throughout the article, we will refer to the MPC/FBP/IF percentage ratio of each SCA: C1 (100/0/0), C2 (50/50/0), C3 (0/50/50), C4 (50/0/50), C5 (75/0/25), C6 (75/25/0), C7 (25/50/25), C8 (25/25/50) and C9 (50/25/25).

All nine formulations contained 49% water, 42.8% vegetable margarine, 0.4% stabiliser mix, 0.6% salt and 0.1% trisodium citrate. The remaining 7.1% was composed of a mixture of MPC, FBP and/or IF in different proportions. All ingredients were weighed using high-precision (±0.0001 g) scales AB304-S (Mettler Toledo, Greifensee, Switzerland) and then mixed in a Thermomix food processor (Thermomix TM31; Vorwerk, Wuppertal, Germany). The fat and water were first blended for 3.5 min at 250 rpm and 50 °C. Then, all powdered ingredients, previously blended, were added to the mixture and blended again for 3 min at 2000 rpm. This method was expected to stimulate the emulsification process. The mixture was pasteurised at 90 °C for 1 min. Finally, the pH was adjusted to 4.6–4.8 using citric acid. This method was expected to stimulate protein aggregation and rennet formation. Then, the mixture was homogenised for 30 s at 300 rpm to give it the right texture. Finally, the cheese analogues were deposited in cylindrical glass jars, covered with lids, labelled and refrigerated at 4 °C for further characterisation. Two batches were elaborated for further analysis. 

#### 2.4.2. Physicochemical Characterisation of Spreadable Hybrid Cheeses

The theoretical nutritional content of the SCAs (protein, crude fat, saturated fat, unsaturated fat, total dietary fibre, carbohydrates, sugar and salt) was calculated using the nutritional content provided by the product data sheets, following Equation (4).
(4)$$m\left(g\right)=m_{1}q_{1}+m_{2}q_{2}+\cdots +m_{n}$$
where *m* = *g* of nutrient per 100 g of SCA, *m_i_* = *g* of ingredient per 100 g of SCA, *q_i_* = (*g* of nutrient per 100 g of ingredient)/100 and *n* = the number of ingredients. 

Energy values were calculated using the Atwater general factors (4 kcal/g for protein and carbohydrates, 9 kcal/g for fat and 2 kcal/g for fibre). The protein-source ingredients constituted 7.1% of the formula and contained similar percentages of protein (between 52 and 70%). The differences between the powdered ingredients were “diluted” in the final product, and changing the ratios of these ingredients did not significantly affect the nutritional profile of the SCAs. As the nutritional profiles of the samples obtained by theoretical estimation were similar, experimental nutritional characterisation was omitted to save resources and reduce costs. 

The moisture content of SCAs was determined by drying an approximately 3 g sample at 105 °C in a drying oven to a constant weight [17] and using Equation (5).
Moisture (%) = (Wet sample weight − dry sample weight)/(wet sample weight) × 100(5)

The pH of cheese samples was determined during the processing and after refrigeration using a pH Meter (pH Meter GLP 21+, Crison Instruments, Alella, Spain).

The colour of cheese samples was measured using the colourimeter in the CIE L* a* b* system, determining the values of L* (lightness), a* (redness/greenness) and b* (yellowness/blueness) colourimetric parameters.

#### 2.4.3. Textural Properties

The texture properties of SCAs were examined using the Spreadability Test, employing a TA.HDplus texturometer (Stable Micro System, Godalming, Surrey, UK) equipped with a Texture Technologies Corporation (TTC) Spreadability Rig and a 5 kg load cell, following the method described by Brighenti, Govindasamy-Lucey [18], with some modifications. 

The TTC spreadability fixture comprises a set of matched Perspex cones, one male and one female, angled at 45 degrees. To ensure accurate measurements, the instrument was calibrated at a test speed of 1 mm/s, with a post-test speed of 1 mm/s, a trigger force of 2 g, and a penetration depth of 23 mm. To prepare for testing, SCA samples were placed inside the female cone, which was then gently pressed down to eliminate air pockets while minimizing structural disruption. Excess sample was removed with a spatula to create a level surface. The test was conducted at a temperature of 18.5 ± 0.5 °C after 48 h of cold storage, which is considered the serving temperature. During the test, the product was squeezed out between the cones, and the force-time curves were recorded.

The maximum force recorded when the probe reached maximum penetration depth was noted as spreadability firmness (Hs, g). The total force required to complete the first phase of the test was measured as spreadability (work of shear, g.mm), which is represented by the area under the curve. During the second phase of the test, the male cone moved in the opposite direction, and the maximum force recorded was termed stickiness (g). The area under the curve corresponding to the work of adhesion (g.mm) was also calculated. Each batch was examined in quadruplicate.

#### 2.4.4. Modelling and Optimisation of Experimental Data

Multiple linear regression analysis was performed to model and predict the moisture, colour (L*, a*, b*) and texture (firmness, spreadability, stickiness and work of adhesion) responses to various protein ingredient combinations. 

Desirability-based optimisation was used to select the samples to be evaluated by the trained panel in sensory analysis. The “desirability” function allows to optimize the experimental design, using the prediction models, selecting the desired maximum, minimum or target value for each response (or parameter) analysed. The function was applied to obtain the samples with the maximum firmness and spreadability (considering closeness to the texture of spreadable dairy cheese) and maximum L* (lightness) values (considering closeness to the colour of dairy cheese). These three parameters were selected for optimisation considering the significance of the regression coefficients obtained in the modeling (that is, the importance of the effect of the combination of ingredients on the parameters) and the relevance of the analysed properties in the desirability of a spreadable cheese-like product. 

#### 2.4.5. Sensory Analysis

A quantitative descriptive analysis (QDA) was performed to assess the sensory characteristics of three of the SCAs.

A semi-trained panel of 8 assessors (5 women and 3 men) was re-trained, during 3 sessions, in the use of a 1–5 intensity scale and in identifying and evaluating sensory characteristics of a commercial dairy spreadable cheese and a commercial plant-based SCA from a Spanish supermarket. The panellists generated the descriptors that they considered important for the descriptive evaluation of spreadable cheese and formulated their definitions in open discussion. The nine attributes generated are described in Table 1. After re-training, a statistical analysis was conducted to confirm the discrimination capacity, repeatability and reproducibility of the panel. 

In QDA evaluation, each panellist examined 5 samples (3 hybrid SCAs and 2 commercial references) presented in random order and labelled with 3-digit random numbers. The panellists rated the attributes of each sample on a 1–5 intensity scale. Three tasting sessions were carried out as replicates.

### 2.5. Statistical Analysis, Modelling and Optimisation of Experimental Data

The experimental design, modelling and optimisation were conducted employing the R-project software v 4.2.1 (R Foundation for Statistical Computing, Vienna, Austria). The package used for the experimental design was “Ternary”. For modelling, the packages employed were “readxl”, “tidyverse”, “mixexp”, “rsm” and “Ternary”, and the polynomial equation used was:(6)$$Y=\beta_{1}X_{1}+\beta_{2}X_{2}+\beta_{3}X_{3}+\beta_{12}X_{1}X_{2}+\beta_{13}X_{1}X_{3}+\beta_{23}X_{2}X_{3}+\beta_{123}X_{1}X_{2}X_{3}$$
where *Y* is the estimated response; $X_{1}$, $X_{2}$ and $X_{3}$ are the ratio of each protein ingredient, and $\beta_{1}$, $\beta_{2}$, $\beta_{3}$, $\beta_{12}$, $\beta_{13}$, $\beta_{23}$ and $\beta_{123}$ are constant coefficients for each linear and nonlinear (interaction) term produced for the prediction models of processing components. The model for each response was obtained based on the fitting quality, the coefficient of determination (R^2^) and the significant level of regression (*p* < 0.05). 

The “desirability” package was used for the optimisation of the experimental design to obtain the maximum firmness and spreadability (considering closeness to the texture of spreadable dairy cheese) and maximum L* (considering closeness to the colour of dairy cheese).

The packages used to analyse the QDA results were “readxl”, “tidyverse”, “ggplot2”, “agricolae” and “fmsb”. For the physicochemical characterisation of protein ingredients and physicochemical, texture and sensory analysis of SCAs, a one-way analysis of variance (ANOVA) and a Tukey HDS test were used to determine pairwise differences between groups. A 5% significance level (*p* < 0.05) was used in all cases.

## 3. Results

### 3.1. Physicochemical Characterisation of Protein Ingredients

Table 2 summarises the functional properties, colour parameters and particle sizes of MPC, FBP and IF. 

The moisture content of MPC, FBP and IF was 5.22, 6.89 and 1.41%, respectively, with significant differences between the three ingredients (*p* < 0.05). This might be due to the different moisture of raw ingredients and the differences in the production processes for powdered ingredients. The water content of MPC coincided with the value given by the manufacturer (<6%). For FBP, the moisture content was slightly lower (10%) and for IF, much lower than reported by the manufacturer (10.74%). In the last case, the moisture might have evaporated during the milling pre-treatment of insect flour.

Swelling capacity was significantly lower for MPC (2.88 mL/g) than for FBP and IF (4.01 and 3.64 mL/g, respectively). This might be due to particle size differences since MPC mainly contained particles of average diameter (d50) of 94.17 µm, significantly larger than in FBP and IF (35.68 and 56.20 µm, respectively). Moreover, d90 (306.50 µm) for MPC was significantly larger than for FBP and IF (130.33 and 161.17 µm, respectively). This means that the MPC ingredient contained coarse particles larger than 300 µm, which could affect its swelling properties [19]. The coarse particles (D[4.3]) in MPC (135.0 µm) were significantly larger than in FBP (54.63 µm) and IF (77.33 µm). The differences in particle size might be due to the different milling procedures used to obtain the powdered ingredients.

The water solubility index was significantly lower for IF (7.62 g/100 g) than for MPC (31.08 g/100 g) and FBP (24.23 g/100 g). The highest value for water-holding capacity was found for MPC (2.69 g/g), followed by IF and FBP (1.64 and 1.35 g/g, respectively). The oil-holding capacity also significantly differed between the ingredients; the highest OHC value was observed for IF (2.10 cm^3^/g), followed by MPC (1.44 cm^3^/g) and FBP (0.72 cm^3^/g). The variation in WHC and OHC values may be due to the differences in protein concentration, conformational characteristics and the degree of interaction with water. An increase in the content of non-polar amino acids strengthens hydrophobic interactions with lipids (and the OHC) and weakens the hydrophilic interactions with water (and the WHC) [19].

The colours of the ingredients differed significantly (*p* < 0.05). L* values closer to 100 indicate light colours, and values closer to 0, dark colours. Positive a* values are associated with a colour close to red, whilst negative values indicate closeness to green. Large positive b* values are observed for colours close to yellow, and small positive b* values indicate closeness to blue. FBP had the highest L* (69.29) and b* (15.13) values and the lowest a* value (−1.92). IF had the lowest L* value (21.53) and the highest a* value (2.44), and the MPC had the lowest b* value (9.63).

### 3.2. Physicochemical Characterisation of the Spreadable Cheese Analogues

Table 3 summarises the colour parameters, the moisture content and the pH of the SCAs.

No significant differences in the moisture content of various SCAs were observed except for samples C3 and C4.

Similarly, there were no significant differences between pH values, except for sample C1, whose pH increased from 4.60 to 5.20 after packaging and refrigeration. The pH remained within the range of 4.6–4.8 for all the other samples. Significant colour differences were detected between some sample groups. The IF content negatively affected the lightness; the samples with 50% IF (C3 and C4) had the lowest L* values. The samples with 25% IF (C5, C7 and C9) had lower L* values than those without IF (C1, C2, C6). In contrast, raising the FBP content increased the L* values of the cheese; samples C2 and C6 had the highest L*. Sample C1, containing only MPC, had an intermediate L* value. Both the MPC and FBP contributed positively to the yellowness (b*) of the SCAs. Sample C1 showed the highest degree of yellowness, and C2 and C6 had higher b* values than the remaining samples. However, IF seemed to counteract the effect of MPC (C4 and C5) and FBP (C3) on yellowness, contributing to redness. C3, C4 and C8 had colours closer to red (a*) because of their high content (50%) of IF. C7 and C5 (25% IF) displayed had lower redness values (they were closer to green). C1 was greener than C2 and C6 because of its 100% MPC content. Samples C2 and C6 contained FBP (50% and 25%, respectively) and had higher greenness values than MPC but also higher levels of yellowness and lightness. C9 had more redness than C7 because (despite the same amount of IF) it contained more MPC and less FBP than C7 (Table 3).

### 3.3. Nutritional Estimates for the Spreadable Cheese Analogues

The theoretical estimation of the nutritional content of SCAs showed that the nine formulations had very similar nutritional profiles. This is probably because the combined protein ingredients represented only 7.1% of the formula, and they contained similar percentages of protein (between 52 and 70%). Thus, the differences between the powdered ingredients were diluted in the final product and changing the ratios of protein ingredients did not significantly affect the nutritional profile of the SCAs (Table 4).

However, it should be noted that adding the FBP affected the fibre, carbohydrate and sugar levels of the samples; FBP contained 14.4% of dietary fibre, 5.3% of starch and 2.1% of sugars. The addition of IF had a slight effect on the fat content since this powder contained 33% of fat (25% UFA and 8% SFA).

### 3.4. Texture Characterisation of the Spreadable Cheese Analogues

The mean values for the textural properties of the different SCAs can be found in Table 5.

Spreadability (work of shear) is defined as the work required to spread a product on an immobile surface; it also reflects the structural breakdown of the product during oral processing [20]. This parameter is considered a good instrumental measure of the spreadability of cream cheeses [21] and other spreadable products [22]. It must be noted that the higher the value of spreadability (work of shear), the lower the spreadability of the product.

All the texture parameters showed similar patterns in all the samples (Figure 3). Moreover, the spreadability (work of shear) was directly proportional to spreadability firmness (Hs), and the stickiness was directly proportional to work of adhesion.

The addition of IF had a significant (*p* < 0.05) effect on the textural properties of SCAs. It caused a reduction in the values of Hs (firmness), spreadability, stickiness and work of adhesion. SCAs with 50% substitution of MCP by IF (C3, C4 and C8) had the lowest firmness (25.5, 36.4 and 36.4 g, respectively), spreadability (70.6, 100.4 and 93.6 g.mm), stickiness (44.7, 61.3 and 65.4 g) and work of adhesion (8.4, 11.3 and 11.9 g.mm) values. Furthermore, sample C3 (with FBP and without MPC) showed significantly lower values than samples C4 and C8 (containing MPC). This indicates that, on its own, MPC increases the firmness of the spreadable cheese more than FBP. However, SCAs with 25% MPC substituted by IF (C5, C7 and C9) showed lower values of firmness, spreadability, stickiness and work of adhesion (46.9, 64.9 and 68.9 g; 121.8, 164.8 and 177.3 g.mm; 79.7, 114.3 and 112.9 g; and 15.0, 22.3 and 23.3 g.mm) than SCAs with no IF, i.e., C1, C2 and C6 (87.1, 137.6 and 87.5; 216.0, 349.9 and 221.3 g.mm; 122.5, 202.1 and 148.9 g; and 28.4, 51.1 and 33.5 g.mm), but higher than for SCAs with 50% IF. Therefore, as the proportion of IF rose, the firmness and stickiness of the SCA decreased, and the spreadability increased.

Within the 0%-IF and 25%-IF sample groups, a synergy effect between MPC and FBP can be observed in samples containing both ingredients. Generally, these samples showed higher values of firmness, spreadability, stickiness and work of adhesion than their counterparts containing MPC but not FBP (C2 and C6 compared with C1, and C7 and C9 compared with C5). This indicates that FBP only had a positive effect on texture when it was combined with MPC, and that there was a synergy between these two ingredients However, as we mentioned before, the IF combined with MPC and FBP (C7, C8 and C9) had an opposite effect on all the texture properties of the SCAs.

### 3.5. Modelling and Optimisation

The results of multiple linear regression analysis for the modeled responses (moisture, colour and texture parameters) are shown in Table 6. The effect of the three different protein sources and their combinations is reflected by the F-value and the corresponding *p*-value. Figure 4 shows ternary diagrams for each response.

Data showed no correlations for the moisture parameter (R^2^ = 0.744); the effect of IF had a low significance even if it was assumed (*p* = 0.028).

For firmness and spreadability, low significance (*p* = 0.001) was observed for the FBP–IF interaction. The effect of this ingredient combination was probably more due to the IF than the FBP content or the interaction between the two, as seen in the texture analysis. Further studies are necessary to examine the individual effect of each ingredient on its own (particularly for the FBP and IF). The loss of stability observed in SCA texture analysis of non-dairy protein sources (FBP and IF) was probably due to the lack of specific interactions between milk proteins and stabilisers such as carrageenan, which affected the final texture of the cream cheese [23,24,25].

A strongly significant correlation was detected between stickiness and MPC content (*p* = 0.001), IF content (*p* < 0.0001) and FBP–IF interaction (*p* = 0.004), and between work of adhesion and FBP content (*p* = 0.002), MPC-FBP interaction (*p* = 0.009) and FBP-IF interaction (*p* < 0.001). FBP and MPC showed opposite effects to IF, as shown in the texture analysis and Figure 4. The FBP increased the stickiness (and, consistently, the work of adhesion) when combined with MPC. The IF significantly reduced the stickiness and work of adhesion when combined with FBP and/or MPC. Therefore, the effect of FBP–IF and MPC–IF interactions on these parameters was probably more related to the IF than to the FBP content, MPC content or the interactions. Further studies are necessary to define the individual effect of each ingredient. This diversity in the functionality of different protein sources could be an opportunity to expand the range of solutions for creating new dairy-free cheese alternatives or reducing their dairy protein content.

Colour parameters also showed correlations with the different protein combinations. MPC content was clearly correlated with the lightness of the SCA (*p* < 0.001). A simultaneous addition of IF increased the redness (*p* < 0.0001) and reduced the yellowness (*p* < 0.001) of the SCA. These changes could be related to the colour of the protein ingredients and the SCA production process.

Finally, the “desirability” function was used to select the samples to be evaluated by the trained panel in sensory analysis. The desirability values were obtained for all samples: C1 (0.821), C2 (0.871), C3 (0.168), C4 (0), C5 (0.410), C6 (0.911), C7 (0.588), C8 (0.084) and C9 (0.5). The aim of this study was to develop a hybrid spreadable cheese (50% or less animal protein) with nutritional value, texture, and flavour similar to those of conventional spreadable cheese. Therefore, the SCAs with more than 50% MPC were not included in the sensory analysis. Accordingly, only the desirability values of SCAs with 50% or less MPC (C2, C3, C4, C7, C8, C9) were considered. A commercial dairy spreadable cheese and a plant-based spreadable cheese analogue were used as QDA reference (the former as texture and appearance reference and the latter as flavour reference). The number of samples that a trained panel can assess simultaneously is usually no more than five to six per session. Therefore, considering the two commercial references, only the three hybrid SCAs with the highest desirability were chosen to be assessed by the trained panel (samples C2, C7 and C9).

### 3.6. Quantitative Descriptive Analysis of Spreadable Cheese Analogues

Table 7 presents the results of the quantitative descriptive analysis performed by the trained panel. Figure 5 shows the sensory profiles of SCAs C2, C7, C9 and the QDA reference for the chosen attributes.

No significant differences in granularity were detected between samples C2 (1) and C9 (1.35) and between C7 (1.6) and C9. However, the panel reported higher granularity scores for C7 than C2 (*p* = 0.00157). This could be explained by the low water solubility index of the IF (7.62%).

Significant differences (*p* ≈ 0) were detected in the creaminess, firmness and adherence of the three analogues. C2 was the creamiest analogue (2.5), followed by C9 (2) and C7 (1.3). C2 was also the sample with the highest firmness (3), with C9 in the second place (2) and C7 (1.25) in the third. Similarly, C2 showed stronger adherence (3.35) than C9 (2.9) and C7 (the weakest adherence, 2.35). These results indicate that these three texture parameters are positively correlated in a directly proportional manner. Sample C2 was significantly less spreadable (4) than samples C7 and C9 (4.85 and 4.7, respectively) (*p* ≈ 0). The spreadability tended to correlate with creaminess, firmness and adherence.

The panel assessed an augmented cheesy flavour in the analogue with the addition of IF; C7 and C9 obtained cheese flavour scores of 3 and 3.4, respectively, and sample C2 showed a significantly lower score (2.6) (*p* ≈ 0). Similarly, the uncharacteristic flavour decreased significantly with the addition of IF (*p* ≈ 0), with an intensity of 3 for sample C2, while samples C7 and C9 had an intensity of 2.65 and 2.35, respectively. Sourness was significantly increased by adding the IF to the formulation (*p* ≈ 0). Sourness intensity scores were 3.7 and 3.85 for C7 and C9, respectively, and 3.1 for C2. Thus, the resemblance to the characteristic cheesy and sour flavour of the dairy cheese increased when IF was added to the formulation.

Sample C2 showed more cheese-characteristic yellowness (score of 4) than samples C7 and C9 (2.95 and 3) (*p* ≈ 0). Thus, the resemblance to the characteristic colour of the dairy cheese decreased in the formulations containing the IF.

The QDA reference was divided in two. For texture (granularity, creamy, firmness, spreadability and adherence) and appearance (characteristic yellow colour) attributes, the developed SCAs were compared to a commercial dairy cheese reference, with the aim of observing if the texture and appearance of the analogues were similar to what would be the ideal of a spreadable cheese. For flavour attributes (cheesy flavour, uncharacteristic flavour and sourness), the developed SCAs were compared to a commercial plant-based cheese analogue reference, with the aim of seeing if the developed analogues had an improved flavour profile compared to that of the commercial plant-based analogues (Figure 5).

Samples C2 and C9 showed granularity values (1 and 1.35) similar to the dairy reference (1); only sample C7 had a slightly higher granularity (1.6) (*p* = 0.00157). All the SCAs showed lower creaminess, firmness and adherence (2.5, 1.3 and 2; 3, 1.25 and 2; 3.35, 2.35 and 2.9, respectively) than the reference sample (5, 4 and 5, respectively). However, their spreadability values were higher (4, 4.85 and 4.7 vs. 3) (*p* ≈ 0). The colours of SCAs differed slightly (4, 2.95 and 3) from the characteristic yellow colour of the QDA reference (5) (*p* ≈ 0).

The flavour profiles showed that the SCAs were more sour (3.1, 3.7 and 3.85 vs. 1), had more cheesy flavour (2.6, 3 and 3.4 vs. 2) and less uncharacteristic flavour (3, 2.65 and 2.35 vs. 4) (*p* ≈ 0) than the commercial plant-based reference. These differences demonstrated the sensory superiority of the new analogues over their commercial counterpart.

## 4. Discussion

### 4.1. Physicochemical Characterisation of Protein Ingredients

Solubility is an important physicochemical and functional property of protein; good solubility favours the formation of emulsions and gels in food products. The IF had a water solubility index of 7.62% (pH 7.38). This result concurs with the data of Zielińska, Karaś [26] obtained using similar assay conditions (for *T. molitor* protein extraction). The authors have reported 3% solubility at pH 5 (minimum solubility value), 97% at pH 11 (maximum solubility value) and 86 and 56% at pH 2 and 3, respectively. A similar trend was observed by Zhao, Vázquez-Gutiérrez [27], with a minimum protein solubility at pH 4 and 5, increasing outside this pH range. These results indicate that the isoelectric point (pI) of *T. molitor* protein is located between pH 4 and 5. Ali [28] has found that the solubility of faba bean protein extract (over the entire pH range) was significantly lower than that of β-lactoglobulin (milk protein). This is in accord with the present study; the solubility observed for MPC (31.08%) was higher than for FBP (24.23%). Abbey and Ibeh [29] have recorded the minimum solubility for brown bean protein at pH 4, which indicates that legume protein pI is also around pH 4.

Water-holding capacity is the ability of a protein matrix to retain, chemically and physically, the maximum amount of water per gram of sample against the force of gravity. In the present study, the IF showed a WHC value of 1.64 g/g. For *T. molitor* whole insect flours, Zielińska, Karaś [26] have recorded a WHC of 1.29 g/g, and Bußler, Rumpold [30], a WHC of 0.8 g/g. These differences in WHC values of mealworm flours may be due to the origin of the insects, the different flour production methods and dissimilar trial conditions [26]. The WHC value obtained for MPC here (2.69 g/g) is lower than those reported by Meena, Singh [31] for MPC60 (5.22 g/g) and by Salunke and Metzger [32] for a 5% (wt/wt) protein solution of MPC (3.48 g/g). The WHC value obtained for FBP (1.35 g/g) is comparable to those reported by Bühler, Dekkers [33] for a faba bean protein concentrate (1.25 g/g) and Sosulski and McCurdy [34] for a 63.3% faba bean protein concentrate at 21 °C (1.03 g/g) and 70 °C (1.53 g/g).

Oil-holding capacity (OHC) indicates the amount of lipids that can be physically absorbed per gram of protein powder [26]. In the present study, IF showed an OHC of 2.10 cm^3^/g. Zielińska, Karaś [26] have obtained an OHC of 1.71 g/g for *T. molitor* flour, and Bußler, Rumpold [30] have reported an OHC of 0.6 g/g. The OHC value obtained here for MPC (1.44 cm^3^/g) is lower than reported by Meena, Singh [31] for an MPC60 (3.38 g/g). The OHC value for FBP (0.72 cm^3^/g) is similar than those reported by Sosulski and McCurdy [34] for a 63.3% faba bean protein concentrate at 21 °C (0.65 g/g) and 70 °C (0.72 g/g).

These results concur with the existing scientific data on food proteins. Protein solubility, WHC and OHC are related to the protein content of the flour; they are affected by the protein size, conformation and amino acid composition. Increasing the number of hydrophobic residues in a protein lowers the solubility and WHC and augments the OHC [26]. Milk proteins are soluble, flexible and amphiphilic. In contrast, pulse proteins have large and compact structures; they are less water-soluble than milk proteins and quite hydrophobic [35]. Compared with the mealworm and faba bean proteins, the relatively large WHC values recorded for MPC reflect the higher content of hydrophilic amino acids in milk proteins and their superior ability to retain water. Generally, legume (peas, lentils) proteins have intermediate WHCs, which can be explained by their high content of water-soluble albumins and salt-water-soluble globulins. However, milk proteins have moderate OHC, whilst some plant proteins (e.g., pea, soy and wheat) are similar to or slightly better at retaining oil than animal proteins [35]. In this study, the OHC of FBP was lower than MPC, suggesting a low hydrophobicity of faba bean proteins compared to other legume proteins. The intermediate WHC value obtained for IF could be caused by the high polar amino acid content of this constituent [26].

### 4.2. Texture Characterisation of the Spreadable Cheese Analogues

Spreadable cream cheese differs structurally from other cheeses by the lack of a compact protein matrix, a relatively high moisture content and a high content of fat. It consists of fat globule clusters interspersed with milk proteins, with a corpuscular structure that contributes to cheese spreadability [36].

For the dairy cheese (C1), a kind of protein matrix formation was achieved by both pH- and heat-induced protein aggregation. The temperature rise during the pasteurisation step could promote a partial denaturation of whey proteins and, therefore, the exposure of hydrophobic residues [36]. Then, during acidification, the pH was decreased to 4.6–4.8, coinciding with the pI of milk proteins, so their net protein charge was zero and protein–protein interactions were at their maximum. This would result in protein aggregation leading to the formation of a casein protein network that entraps water and fat.

*T. molitor* and legume proteins have a pI similar to that of milk protein [11,12,37]. Thus, the present study could assess how these proteins correctly replicate the casein aggregation at a pH of 4.6–4.8 or interact positively with the casein matrix. However, the texture analysis demonstrated that faba bean protein on its own had poorer gelation properties than milk protein, and that *T. molitor* protein had weaker gelation ability than milk and faba bean proteins.

The lower firmness of the SCAs with 50% and 25% IF was probably due to insect protein affecting protein–protein interactions in the casein matrix, weakening them and loosening the matrix. Similar results have been obtained for hybrid meat products containing insect and pork proteins, which were unable to aggregate, decreasing the gel strength [38]. This would mean that the insect protein used in this study had a low functionality. Incorporating non-functional proteins might also disrupt the protein interaction in the casein matrix, reducing firmness. This is in accord with Lee, Huss [39] who observed reduced firmness in processed cheese spreads with added non-functional whey protein.

The increased firmness and stickiness observed in SCAs combining MPC and FBP compared to samples containing MPC but not FBP and samples containing FBP and IF but not MPC indicate a synergistic effect between MPC and FBP. The synergy could be explained by the effect of the fibre and starch in FBP, which act as thickeners by increasing the viscosity of the continuous phase. The fibre in the FBP could improve the gel strength by forming a stable network structure stabilising the water phase, and by the filling effect [40]. Zhuang, Jiang [41] have found that the insoluble dietary fibre from sugarcane improves the myofibrillar protein gelation by stabilising moisture, thanks to its high WHC. Moreover, the starch content of FBP would help to immobilise water in the matrix and increase the hardness of the SCA [8,42]. Given the assay conditions, starch probably acted as a filler material without network formation [43]. However, the interaction between the dairy and faba bean protein during thermal processing could produce a synergistic effect, improving the textural properties. Similar results have been reported by Chihi, Mession [44], obtaining “mixed” pea–whey protein aggregates, and by Roesch and Corredig [45], making soy and whey protein aggregates.

Therefore, faba bean protein itself is not able to form the necessary protein network as casein can, as Ferawati, Hefni [8] indicate. However, in the presence of a protein network (samples with MPC), as FBP content increases, fibre and starch would retain more water within the milk protein network, increasing the viscosity and consistency of the gel.

Yi, Lakemond [46] evaluated the gel-forming ability of *T. molitor* protein concentrate in a pH range of 3 to 10. In their study, the gel was formed at pH 7 and 10, at 30% (*w/v*) protein concentration and 61.7 ± 1.1 °C. This means that at the pH range of 7–10, heat promoted protein denaturation, aggregation and gelation. The pH used in the present study was not suitable for gelation. This may be an explanation for the texture of samples containing IF.

### 4.3. Sensory Analysis of the Spreadable Cheese Analogues

To the best of our knowledge, there is no data on using faba bean protein to make plant-based SCAs. Neither cheese analogues containing insect protein nor combined insect and pulse protein have been produced.

The colour of samples containing IF, described by the sensory panel as different from the characteristic yellow of traditional cheese, is consistent with the colourimetric analysis results. The products with IF were darker and further from green and yellow than those without this ingredient. This result is similar to the data reported for a *T. molitor* larva-based milk alternative developed by Tello, Aganovic [5], who obtained a beige, opaque prototype.

The OHC is related to taste and texture, the desirable properties affecting flavour retention and tenderness, making food more palatable [12]. The high OHC of the IF ingredient may increase the retention of fat by the protein in samples with IF compared to those without, hindering water retention and the acquisition of a thick texture. This could explain the sensory profiles obtained for samples C7 and C9. Sample C2 (MPC/FBP/IF ratio: 0.5/0.5/0) was creamier, firmer and more adherent than C7 (0.25/0.5/0.25) and C9 (0.5/0.25/0.25). Thus, samples containing a larger proportion of FBP and MPC have more desirable texture profiles than their counterparts.

Dairy cream cheese is a soft, unmatured cheese, spreadable at room or cooling temperature (7–20 °C), with a white to light yellow colour, smooth texture, slight lactic acidity and buttery taste and smell [47]. The three SCAs analysed here were less creamy, less firm, with weaker adherence than the dairy reference, and were more spreadable (Figure 5). The panellists considered the high spreadability of samples C7 and C9 unsuitable for an SCA as it was associated with a lack of firmness and excessive fluidity. Future work aimed to improve the texture of SCAs could involve increasing the proportion of solids (protein, starch and fibre). However, the SCAs were more sour than the plant-based reference, which is a desirable attribute in dairy cream cheese [47]. This was probably due to the addition of citric acid and IF. The panel noted a reduction in the uncharacteristic flavour of SCAs compared to the commercial plant-based reference. This can be attributed to the presence of IF, which partially masks the buttery flavour and provides an umami flavour reminiscent of old cheese. The beany flavour found in many legume-based cheese analogues [48,49] and in cheese-like products enriched with soy protein [2] was also absent. From a sensory point of view, these differences make the new analogues superior to commercial plant-based products.

### 4.4. Nutritional Profile of the Spreadable Cheese Analogues

A search of the Mintel database (December 2022) was carried out to analyse the nutritional composition of vegan cheese analogues launched on the European market between 2017 and 2022 (search filters: “spreadable cheese”, “vegan/no animal ingredients”, “unflavoured/plain”, “Europe” and “2017–2022”). Most commercial plant-based spreadable cheese analogues available in Europe consist of water, oils rich in saturated fats, starch, and stabilisers. Accordingly, these products are very poor in protein, so they do not have the same nutritional quality as conventional cheese [6,7,8]. Compared to these commercial analogues, the SCAs developed in this study had an improved nutritional profile, resembling a dairy cheese profile (mainly moisture, fat and protein) [21]. They had higher protein content, lower content of saturated fats (substituted by unsaturated fats), reduced content of carbohydrates (mainly composed of starch) and lower sugar content than the cheese analogues described in the Mintel database (Figure 6).

## 5. Conclusions

The study demonstrates that the inclusion of alternative protein sources can be effective as a strategy to reduce animal protein content in hybrid product formulations. The two alternative protein sources studied, the faba bean flour (FBP) and the insect flour (IF, *T. molitor*), can be combined with a dairy protein source (MPC) to obtain a hybrid spreadable cheese. Its nutritional and sensory properties are similar or even superior to those of the 100% dairy and 100% plant-based products. The FBP flour improved the texture (increasing firmness and stickiness and decreasing spreadability), but only when combined with MPC (synergistic effect). Sensory analysis showed that hybrid SCAs (≤50% MPC) C2, C7 and C9 had a more characteristic cheesy flavour than the commercial plant-based reference, and sample C2 had a texture profile similar to the dairy reference. Samples containing IF (C7 and C9) showed a better flavour profile than that without IF (C2). The SCAs had higher protein and lower saturated fat, starch and sugar content than commercial analogues.

Future work should include microstructure analysis of SCAs to improve our knowledge of the alternative protein functionality and the effects of such ingredients on the casein matrix and the SCA texture. Furthermore, physicochemical characterisation of the combinations of protein components to be used in the SCA formulations would help in understanding the interactions between the ingredients.

## Figures and Tables

**Figure 1 foods-12-01522-f001:**
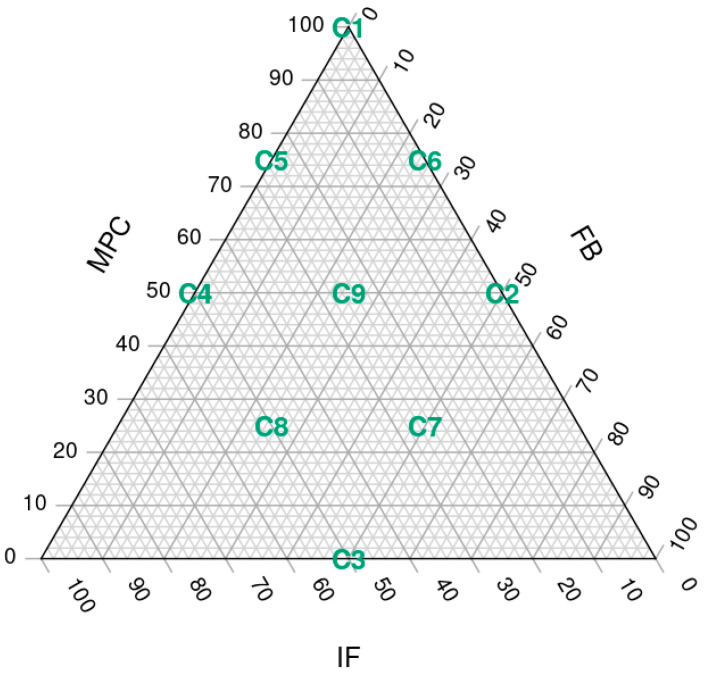
Simplex lattice design plot of ingredient proportions representing 7.1% of spreadable cheese analogue formula, with MPC replacements adding up to 100%. The samples, labelled from C1 to C9, contain different ratios of milk protein concentrate, faba bean protein concentrate and insect flour (MPC/FBP/IF): C1 (100/0/0), C2 (50/50/0), C3 (0/50/50), C4 (50/0/50), C5 (75/0/25), C6 (75/25/0), C7 (25/50/25), C8 (25/25/50) and C9 (50/25/25).

**Figure 2 foods-12-01522-f002:**
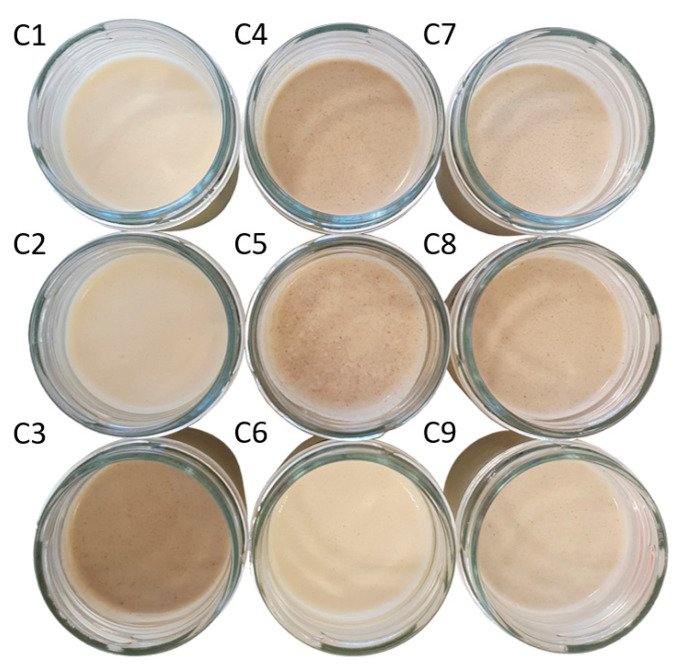
Images of the nine spreadable cheese analogues used in the experiments. The samples, labelled from C1 to C9, contain different ratios of milk protein concentrate, faba bean protein concentrate and insect flour (MPC/FBP/IF): C1 (100/0/0), C2 (50/50/0), C3 (0/50/50), C4 (50/0/50), C5 (75/0/25), C6 (75/25/0), C7 (25/50/25), C8 (25/25/50) and C9 (50/25/25).

**Figure 3 foods-12-01522-f003:**
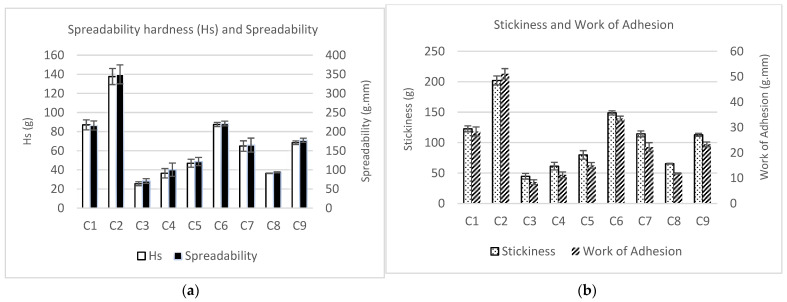
Instrumental texture analysis of the nine spreadable cheese analogues: firmness and spreadability (**a**) and stickiness and work of adhesion (**b**). The samples, labelled from C1 to C9, contain different proportions of milk protein concentrate, faba bean protein concentrate and insect flour (MPC/FBP/IF): C1 (100/0/0), C2 (50/50/0), C3 (0/50/50), C4 (50/0/50), C5 (75/0/25), C6 (75/25/0), C7 (25/50/25), C8 (25/25/50) and C9 (50/25/25).

**Figure 4 foods-12-01522-f004:**
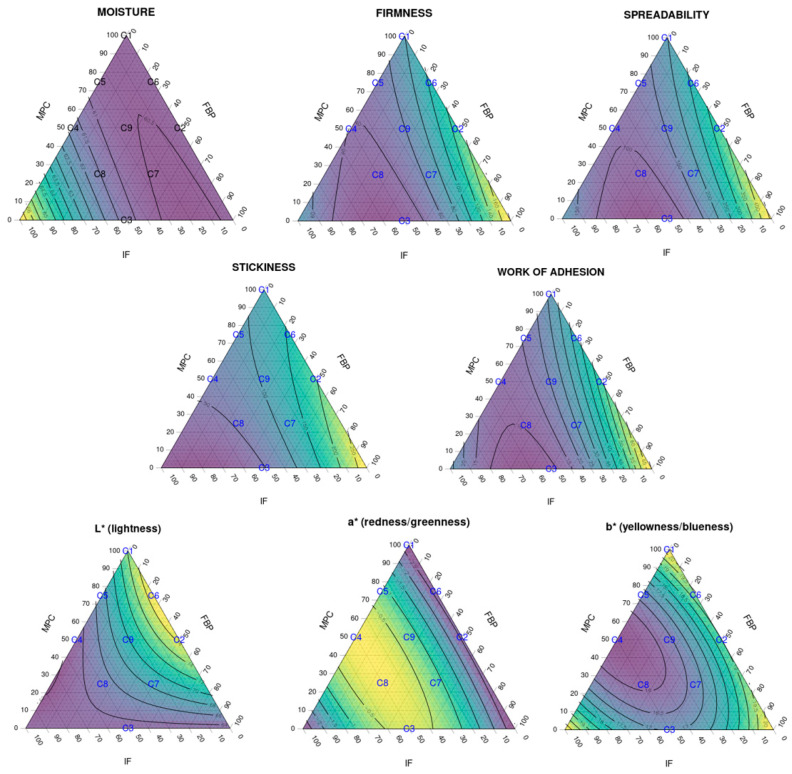
Ternary diagrams showing the effect of the three different protein sources and their combinations to the analysed responses: spreadability, stickiness, work of adhesion and L* a* b* colour parameters (R^2^ > 90% for all responses except for moisture, R^2^ = 0.744). The samples, labelled from C1 to C9, contain different proportions of milk protein concentrate, faba bean protein concentrate and insect flour (MPC/FBP/IF): C1 (100/0/0), C2 (50/50/0), C3 (0/50/50), C4 (50/0/50), C5 (75/0/25), C6 (75/25/0), C7 (25/50/25), C8 (25/25/50) and C9 (50/25/25).

**Figure 5 foods-12-01522-f005:**
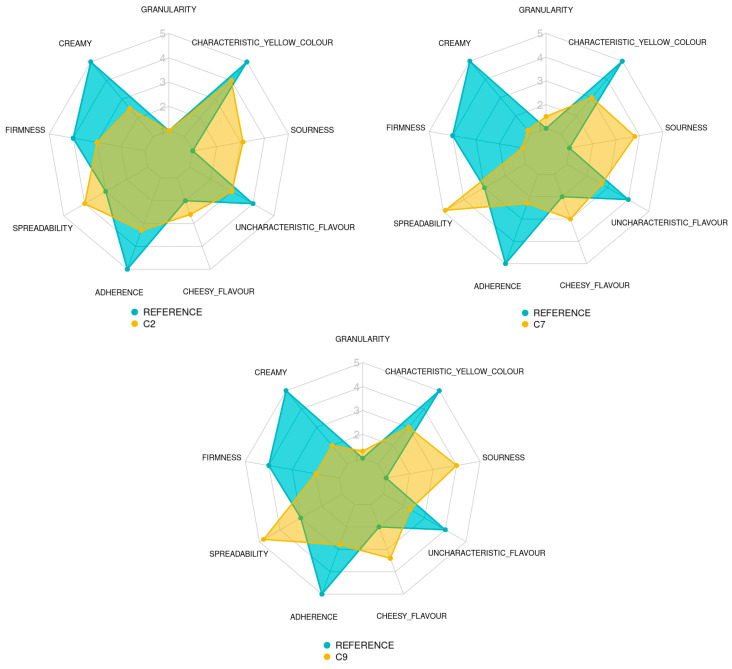
Spider charts showing the sensory results obtained by the trained panel for each attribute of the 3 spreadable cheese analogues and the QDA reference (a commercial dairy cheese and plant-based cheese). MPC/FBP/IF ratio: C2 (50/50/0), C7 (25/50/25) and C9 (50/25/25).

**Figure 6 foods-12-01522-f006:**
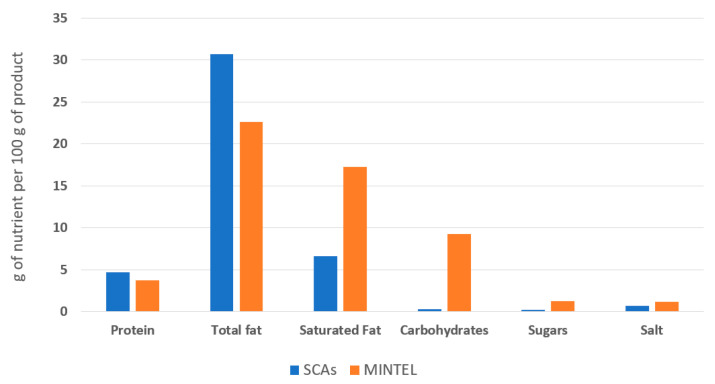
The average nutritional profile of the commercial vegan spreadable cheese analogues on the European market between 2017 and 2022 (based on a search of the Mintel database) and the spreadable cheese analogues developed in the present study (SCAs).

**Table 1 foods-12-01522-t001:** Sensory descriptors for the spreadable cheese analogues.

	Sensorial Attribute	Definition
Texture by mouthfeel	Granularity	Presence of fine particles or granules. 1 = Low-as smooth cheese; 5 = High-as grainy cheese.
	Creaminess	Creamy feeling of fullness in the mouth. 1 = not creamy; 5 = Very creamy.
	Firmness	Effort needed to compress and break down the cheese when pressed to the roof of the mouth. 1 = Smooth; 5 = Firm.
Texture by hand	Spreadability	Ease with which cheese can be spread on a cracker. 1 = little spreadable, it required much effort to spread; 5 = very spreadable, less effort is required to spread.
	Adhesion	Ease with which cheese sticks to bread or knife. 1 = Low adhesion 5 = High adhesion.
Taste	Sourness	The basic taste, perceived on the tongue, stimulated by acids such as critic acid. Pungent acidic aroma/flavour resembling sour cream. 1 = little sour; 5 = very sour.
	Cheesy flavour	Characteristic flavour of the cheese. 1 = little cheesy flavour; 5 = lots of cheese flavour.
	Uncharacteristic flavour	Beany flavour or buttery flavour. 1 = non uncharacteristic flavours; 5 = strong uncharacteristic flavours.
Appearance	Characteristic yellow colour	Pale yellow, such as cheese colour. 1 = very different colour; 5 = characteristic pale yellow colour.

**Table 2 foods-12-01522-t002:** Physicochemical characteristics of milk protein concentrate (MPC), faba bean protein concentrate (FBP) and insect flour (IF). The entries show means ± standard deviations, n = 6.

Functional Properties	MPC	FBP	IF	*p*-Value
Moisture (%)	5.22 ^a^ ± 0.25	6.89 ^b^ ± 0.03	1.41 ^c^ ± 0.03	<0.001 ***
Swelling capacity (mL/g)	2.88 ^a^ ± 0.09	4.01 ^b^ ± 0.11	3.64 ^b^ ± 0.37	0.0338 *
Water solubility index (%)	31.08 ^a^ ± 7.05	24.23 ^a^ ± 0.83	7.62 ^b^ ± 1.26	0.0234 *
Water-holding capacity (g/g)	2.69 ^a^ ± 0.09	1.35 ^b^ ± 0.08	1.64 ^c^ ± 0.08	0.0011 *
Oil-holding capacity (cm^3^/g)	1.44 ^a^ ± 0.05	0.72 ^b^ ± 0.04	2.10 ^c^ ± 0.02	<0.001 ***
**Colour parameters**				
L*	69.29 ^a^ ± 0.02	79.92 ^b^ ± 0.01	21.53 ^c^ ± 0.01	<0.001 ***
a*	−1.80 ^a^ ± 0.02	−1.92 ^b^ ± 0.02	2.44 ^c^ ± 0.06	<0.001 ***
b*	9.63 ^a^ ± 0.04	15.13 ^b^ ± 0.02	12.70 ^c^ ± 0.01	<0.001 ***
**Particle size characteristics**				
d10 (µm)	25.90 ^a^ ± 0.31	6.45 ^b^ ± 0.19	7.11 ^b^ ± 0.98	<0.001 ***
d50 (µm)	94.17 ^a^ ± 3.44	35.68 ^b^ ± 2.40	56.20 ^c^ ± 7.45	<0.001 ***
d90 (µm)	306.50 ^a^ ± 24.12	130.33 ^b^ ± 7.39	161.17 ^c^ ± 27.33	<0.001 ***
D[4.3] (µm)	135.0 ^a^ ± 8.22	54.63 ^b^ ± 3.16	77.33 ^c^ ± 13.24	<0.001 ***
D[3.2] (µm)	52.18 ^a^ ± 0.73	14.95 ^b^ ± 0.47	14.72 ^b^ ± 1.56	<0.001 ***
Specific Surface area	115.0 ^a^ ± 1.57	401.88 ^b^ ± 12.49	411.42 ^b^ ± 43.54	<0.001 ***

Means with different superscript letters in the same row are significantly different according to Tukey’s test (*p* < 0.05). Significance codes for *p*-value: ‘***’ (*p* < 0.001); ‘*’ (0.001 < *p* < 0.05).

**Table 3 foods-12-01522-t003:** Physicochemical characterisation of the spreadable cheese analogues. Values show means ± standard deviations, n = 36 for colour parameters and n = 18 for pH and moisture content.

	Ratio MPC/FBC/IF	Moisture (%)	pH	L*	a*	b*
C1	100/0/0	60.72 ^a^ ± 0.36	5.20 ^a^ ± 0.20	74.42 ^a^ ± 0.26	−2.99 ^a^ ± 0.03	20.32 ^a^ ± 0.69
C2	50/50/0	60.72 ^a^ ± 0.16	4.60 ^b^ ± 0.28	76.81 ^b^ ± 0.17	−2.57 ^b^ ± 0.06	19.48 ^b^ ± 0.19
C3	0/50/50	61.47 ^b^ ± 0.05	4.41 ^b^ ± 0.06	62.57 ^c^ ± 0.43	−0.25 ^c^ ± 0.19	17.23 ^c^ ± 0.79
C4	50/0/50	62.15 ^c^ ± 0.28	4.55 ^b^ ± 0.13	63.46 ^c^ ± 0.61	−0.15 ^c^ ± 0.17	15.93 ^e^ ± 0.60
C5	75/0/25	60.55 ^a^ ± 0.16	4.63 ^b^ ± 0.09	67.43 ^d^ ± 0.22	−0.86 ^de^ ± 0.14	16.68 ^cd^ ± 0.21
C6	75/25/0	60.66 ^a^ ± 0.06	4.56 ^b^ ± 0.12	76.84 ^b^ ± 0.28	−2.55 ^b^ ± 0.06	19.18 ^a^ ± 0.38
C7	25/50/25	60.61 ^a^ ± 0.07	4.43 ^b^ ± 0.04	70.06 ^e^ ± 0.98	−1.04 ^d^ ± 0.12	16.40 ^a^ ± 0.14
C8	25/25/50	60.93 ^a^ ± 0.02	4.46 ^b^ ± 0.13	65.27 ^f^ ± 0.29	−0.21 ^c^ ± 0.11	15.92 ^a^ ± 0.07
C9	50/25/25	60.59 ^a^ ± 0.01	4.55 ^b^ ± 0.07	68.08 ^d^ ± 1.81	−0.79 ^e^ ± 0.32	15.95 ^a^ ± 0.69

Means with different superscript letters in the same column are significantly different according to Tukey’s test (*p* < 0.05).

**Table 4 foods-12-01522-t004:** Theoretical nutritional profiles of the nine spreadable cheese analogues. The samples, labelled from C1 to C9, contain different proportions of milk protein concentrate, faba bean protein concentrate and insect flour (MPC/FBP/IF): C1 (100/0/0), C2 (50/50/0), C3 (0/50/50), C4 (50/0/50), C5 (75/0/25), C6 (75/25/0), C7 (25/50/25), C8 (25/25/50) and C9 (50/25/25).

	Energy (kJ)	Energy (Kcal)	Protein (g)	Total Fat (g)	SFA (g)	UFA (g)	CH (g)	Sugars (g)	Total Fibre (g)	Salt (g)
C1	1204.88	292.35	5.18	30.08	6.51	23.15	0.21	0.21	0.00	0.69
C2	1208.05	293.17	4.83	30.14	6.49	23.20	0.40	0.29	0.51	0.69
C3	1239.41	300.89	4.20	31.25	6.73	24.06	0.40	0.29	0.64	0.70
C4	1236.24	300.07	4.55	31.19	6.75	24.01	0.21	0.21	0.13	0.70
C5	1220.56	296.21	4.87	30.64	6.63	23.58	0.21	0.21	0.07	0.69
C6	1206.47	292.76	5.01	30.11	6.50	23.17	0.31	0.25	0.26	0.69
C7	1223.73	297.03	4.51	30.69	6.61	23.63	0.40	0.29	0.58	0.69
C8	1237.82	300.48	4.37	31.22	6.74	24.04	0.31	0.25	0.39	0.70
C9	1222.15	296.62	4.69	30.66	6.62	23.61	0.31	0.25	0.32	0.69

**Table 5 foods-12-01522-t005:** Texture properties of the nine spreadable cheese analogues. Values show means ± standard deviations, n = 36.

	Ratio MPC/FBP/IF	Hs (g)	Spreadability (g.mm)	Stickiness (g)	Work of Adhesion (g.mm)
C1	100/0/0	87.1 ^b^ ± 5.2	216.0 ^c^ ± 12.0	122.5 ^b^ ± 4.9	28.4 ^c^ ± 1.8
C2	50/50/0	137.6 ^a^ ± 8.4	349.9 ^a^ ± 24.9	202.1 ^a^ ± 7.3	51.1 ^a^ ± 2.1
C3	0/50/50	25.5 ^e^ ± 2.1	70.6 ^e^ ± 6.2	44.7 ^e^ ± 4.6	8.4 ^g^ ± 0.9
C4	50/0/50	36.4 ^d^ ± 5.0	100.4 ^d^ ± 17.3	61.3 ^d^ ± 6.3	11.3 ^f^ ± 1.2
C5	75/0/25	46.9 ^d^ ± 4.2	121.8 ^d^ ± 10.6	79.7 ^c^ ± 7.1	15.0 ^e^ ± 1.1
C6	75/25/0	87.5 ^b^ ± 2.2	221.3 ^b^ ± 6.2	148.9 ^a^ ± 3.4	33.5 ^b^ ± 0.9
C7	25/50/25	64.9 ^c^ ± 5.5	164.8 ^c^ ± 18.3	114.3 ^b^ ± 4.8	22.3 ^d^ ± 1.7
C8	25/25/50	36.4 ^d^ ± 0.3	93.6 ^de^ ± 1.2	65.4 ^d^ ± 0.8	11.9 ^f^ ± 0.2
C9	50/25/25	68.4 ^c^ ± 1.9	177.3 ^b^ ± 5.5	112.9 ^b^ ± 2.5	23.3 ^d^ ± 0.9

Means with different superscript letters in the same column are significantly different according to Tukey’s test (*p* < 0.05).

**Table 6 foods-12-01522-t006:** Summary of statistics for the multiple linear regression, showing R-squared fit, F-value, and *p*-value for the model and each independent variable and their interactions.

	Moisture	Firmness (g)	Spreadability (g.mm)	Stickiness (g)	Work of Adhesion (g.mm)	L*	a*	b*
R^2^	0.744	0.945	0.944	0.977	0.969	0.973	0.991	0.965
F	4.848	28.455	27.900	71.150	51.842	60.260	176.372	46.122
Pr > F	0.061	0.001	0.002	0.000	0.000	0.000	<0.000	0.000
MPC	-	-	-	43.568	-	68.581	-	-
	-	-	-	0.001	-	0.000	-	-
FBP	-	15.134	16.434	-	32.981	-	-	-
	-	0.012	0.010	-	0.002	-	-	-
IF	9.387	-	-	148.689	-	-	-	-
	0.028	-	-	<0.0001	-	-	-	-
MPC–FBP	-	-	-	-	-	42.280	3.457	9.913
	-	-	-	-	-	0.001	0.122	0.025
MPC–IF	4.646	10.359	8.805	-	16.805	21.165	291.024	109.041
	0.084	0.024	0.031	-	0.009	0.006	<0.0001	0.000
FBP–IF	6.007	58.259	58.657	25.839	105.231	-	268.403	58.037
	0.058	0.001	0.001	0.004	0.000	-	<0.0001	0.001
MPC–FBP–IF	-	-	-	-	-	-	-	-
	-	-	-	-	-	-	-	-

(-) No interactions were found with the available data.

**Table 7 foods-12-01522-t007:** Sensory characterisation of the spreadable cheese analogues. Values are shown as means ± standard deviations. F-values and *p*-values for the product from one-way ANOVA on all descriptors are also indicated.

Descriptors	Reference	C2	C7	C9	F-Value	*p*-Value
Granularity	1.0 ^a^ ± 0.0	1.0 ^a^ ± 0.0	1.60 ^b^ ± 0.6	1.35 ^ab^ ± 0.5	6.26	0.00157 **
Creaminess	5.0 ^a^ ± 0.0	2.5 ^b^ ± 0.5	1.3 ^d^ ± 0.4	2.0 ^c^ ± 0.0	259.30	<2 × 10^−16^ ***
Firmness	4.0 ^a^ ± 0.0	3.0 ^b^ ± 0.0	1.25 ^d^ ± 0.3	2.0 ^c^ ± 0.0	825.0	<2 × 10^−16^ ***
Spreadability	3.0 ^a^ ± 0.0	4.0 ^b^ ± 0.0	4.85 ^c^ ± 0.3	4.7 ^c^ ± 0.4	120.70	<2 × 10^−16^ ***
Adherence	5.0 ^a^ ± 0.0	3.35 ^b^ ± 0.4	2.35 ^d^ ± 0.4	2.9 ^c^ ± 0.5	94.91	<2 × 10^−16^ ***
Cheesy flavour	2.0 ^a^ ± 0.0	2.6 ^b^ ± 0.5	3.0 ^bc^ ± 0.2	3.4 ^c^ ± 0.5	26.75	2.82 × 10^−9^ ***
Uncharacteristic flavour	4.0 ^a^ ± 0.0	3.0 ^b^ ± 0.0	2.65 ^bc^ ± 0.3	2.35 ^c^ ± 0.5	45.78	2.26 × 10^−12^ ***
Sourness	1.0 ^a^ ± 0.0	3.1 ^b^ ± 0.3	3.7 ^c^ ± 0.4	3.85 ^c^ ± 0.2	246.70	<2 × 10^−16^ ***
Characteristic colour	5.0 ^a^ ± 0.0	4.0 ^b^ ± 0.0	2.95 ^c^ ± 0.2	3.0 ^c^ ± 0.0	1508.0	<2 × 10^−16^ ***

Means with different superscript letters in the same row are significantly different according to Tukey’s test (*p* < 0.05). Significance codes for *p*-value: ‘***’ (*p* ≈ 0), ‘**’ (*p* < 0.001).

## Data Availability

The data presented in this study are available on request from the corresponding author.

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
