# Peer review of "Hybrid Spreadable Cheese Analogues with Faba Bean and Mealworm (Tenebrio molitor) Flours: Optimisation Using Desirability-Based Mixture Design"

_foods, 2023, doi:10.3390/foods12071522_

Round 1
Reviewer 1 Report
The manuscript entitled Hybrid spreadable cheese analogues with faba bean and Tenebrio molitor flours: optimisation using desirability-based mixture design is related to an exciting topic of high interest for food and guaranteeing food safety. The topic is new and can be highly valued by engineers and food businesspeople.
However, despite the merits pointed out, the presentation of the work and the quality of the results shown is low and should be improved and rewritten in a better way to reach the high standards that Foods magazine has achieved.
Many formal shortcomings are pointed out directly in the document for review and correction by the authors. Generally, the most common formal deficiencies were related to the document format, which differs from the template offered by Foods journal.
1) The Abstract exceeds 200 words, the maximum value accepted by the journal.
2) The Tables and Figures do not comply with the suggested format.
3) The "Discussion" section is not in the suggested format.
4) The equations shown (from 1 to 6) in work must be elaborated using the equation editor of MS Word.
5) The units of measurement of the physical-chemical properties shown must be carefully reviewed. Force must be expressed in grams-force (gf), not grams (g). Moreover, the work is expressed in gf x mm, not g.mm.
Additionally, although great experimental work is observed, later, all the information obtained from the experiments was not used adequately in the best way.
1) For example, despite the numerous answers obtained from the nine experimental variants of the analysed mixtures (C1 to C9), it does not seem sufficiently explained why it was decided to include only three in the desirability function (two from the sensory analysis: firmness and spreadability, and a physical-chemical property related to colour (L*)) of the multiple responses obtained from the physical-chemical and sensory properties measured for each mixture variant.
2) Why it was decided to explore a linear model with interactions of the type: Y = β1 X1 + β2 X2 + β3 X3 + β12 X1X2 + β13 X1 X3 + β23 X2 X3 + β123 X1 X2 X3, (equation which is not numbered in the text) instead of a quadratic one? What are the factors X1, X2 and X3? Perhaps the mass fractions or their percentages of MPC, FBP and IF?
3) Why was the codified equation of the desirability function obtained from the three previously mentioned answers not presented explicitly?
4) What were the assumptions for optimising the desirability function?
5) Did the optimum values obtained coincide with the conditions of the mixtures C2, C7 and C9?
6) Why were additional confirmatory experiments not performed to demonstrate the validity of the model's predictions?
For all of the above, the work should be thoroughly reviewed to be considered for publication in the Foods journal.

Author Response
REVIEWER 1
The manuscript entitled Hybrid spreadable cheese analogues with faba bean and Tenebrio molitor flours: optimisation using desirability-based mixture design is related to an exciting topic of high interest for food and guaranteeing food safety. The topic is new and can be highly valued by engineers and food businesspeople.
However, despite the merits pointed out, the presentation of the work and the quality of the results shown is low and should be improved and rewritten in a better way to reach the high standards that Foods magazine has achieved.
Many formal shortcomings are pointed out directly in the document for review and correction by the authors.
- Tenebrio molitor has been written in italics
- The Abstract has 225 word and according to editorial standard it should have a maximum of 200 words.
Abstract has been reduced.
- Line 59: Pulses (beans?)
Pulses in general are interesting protein sources. Then we specify why faba bean is of great interest.
- Line 93: citation mistakes (??) Talens, Llorente [14] and Talens, Lago [15]. Substitute “,” by “&”.
Substituted
- Line 111: (SFA) and (UFA).
corrected
- Line 114: Milk protein concentrate (MPC)
(MPC) has been added
- Line 220: 5-kg. Remove “-”
Removed
- Line 225: indicate force as g-force, not g alone
The force is expressed as “g” not “gf”. When talking about firmness in texture, the force is expressed in grams, it is understood that it is grams-force.
The following articles can be consulted as an example:
- https://doi.org/10.3168/jds.2020-18160
- https://doi.org/10.1016/j.ijgfs.2021.100376
- https://doi.org/10.1016/j.foodres.2018.02.069
- http://dx.doi.org/10.1016/j.lwt.2017.08.004
- http://dx.doi.org/10.1016/j.meatsci.2013.10.031
- Line 243: check the units g.mm
Examples where units of work in texture measurements are g.mm:
- http://dx.doi.org/10.1016/j.meatsci.2013.10.031
We understand that it can be named as g⋅mm, as g.mm as g mm:
- https://doi.org/10.1016/j.ijbiomac.2022.10.146 (they refer to it as g⋅mm and as g.mm)
- http://dx.doi.org/10.1016/j.meatsci.2013.10.031 (they refer to it as g.mm and as g mm)
- Line 268: “square regression coefficient of determination”
We have checked if “coefficient of determination” is correct for R2, and yes, it is.
- Line 300: how many replicates were analysed? N=?
It is now indicated in the text. We analysed two replicates per sample, so 2 replicates per 3 samples (protein ingredients), n=6.
- Table 2: It is not necessary the p-value column
p-value indicates the magnitude of the difference. In our opinion it does show relevant information. In previous work, reviewers have suggested to include it.
- TABLE 2: rewrite the table according to the format suggested in the template document of the Food journal
Rewritten
- TABLE 3: rewrite the table according to the format suggested in the template document of the Food journal
Rewritten
- Table 3: how many replicates were executed? If n= 2 or 3, it would be better put the actual values in the table instead mean +/- sd.
The measurements were done in triplicate, so the mean and sd are used for the results and discussion section.
- TABLE 4: rewrite the table according to the format suggested in the template document of the Food journal
Rewritten
- TABLE 5: rewrite the table according to the format suggested in the template document of the Food journal
Rewritten
- TABLE 5: Please, notify the number of replicate. Also, it is much better present the experimental values of each replicate instead the value of mean +/- sd.
The number of replicates is also indicated in Materials and Methods section.
- Figure 3: Rewrite the figure and the figure caption text according to the format suggested in the template document of the Foods journal.
Figure caption re-written
- Line 404-407: This explanation looks better put in the Materials and Methods section.
We prefer to leave the explanation in this section as it helps understanding the flow of the discussion
- TABLE 6: rewrite the table according to the format suggested in the template document of the Food journal
Rewritten
- TABLE 6: ??????
We have added a (-) sign to fill the empty cells and included the explanation in the legend: (-) No interactions were found with the available data.
TABLE 7: rewrite the table according to the format suggested in the template document of the Food journal
Rewritten
- TABLE 7: F-value and p-value columns are unnecessary.
F-value and p-value indicates the magnitude of the difference. In our opinion it does show relevant information. In previous work, reviewers have suggested to include it.
Generally, the most common formal deficiencies were related to the document format, which differs from the template offered by Foods journal.
1) The Abstract exceeds 200 words, the maximum value accepted by the journal.
Abstract reduced
2) The Tables and Figures do not comply with the suggested format.
Reviewed
3) The "Discussion" section is not in the suggested format.
Reviewed
4) The equations shown (from 1 to 6) in work must be elaborated using the equation editor of MS Word.
We have problems to convert words in equations, so in previous works we introduced the equations in the actual format and then editors changed them if necessary.
5) The units of measurement of the physical-chemical properties shown must be carefully reviewed. Force must be expressed in grams-force (gf), not grams (g). Moreover, the work is expressed in gf x mm, not g.mm.
The force is expressed as “g” not “gf”. When talking about firmness in texture, the force is expressed in grams, it is understood that it is grams-force.
The following articles can be consulted as an example:
- https://doi.org/10.3168/jds.2020-18160
- https://doi.org/10.1016/j.ijgfs.2021.100376
- https://doi.org/10.1016/j.foodres.2018.02.069
- http://dx.doi.org/10.1016/j.lwt.2017.08.004
- http://dx.doi.org/10.1016/j.meatsci.2013.10.031
Examples where units of work in texture measurements are g.mm:
- http://dx.doi.org/10.1016/j.meatsci.2013.10.031
We understand that it can be named as g⋅mm, as g.mm as g mm:
- https://doi.org/10.1016/j.ijbiomac.2022.10.146 (they refer to it as g⋅mm and as g.mm)
- http://dx.doi.org/10.1016/j.meatsci.2013.10.031 (they refer to it as g.mm and as g mm)
Additionally, although great experimental work is observed, later, all the information obtained from the experiments was not used adequately in the best way.
1) For example, despite the numerous answers obtained from the nine experimental variants of the analysed mixtures (C1 to C9), it does not seem sufficiently explained why it was decided to include only three in the desirability function (two from the sensory analysis: firmness and spreadability, and a physical-chemical property related to colour (L*)) of the multiple responses obtained from the physical-chemical and sensory properties measured for each mixture variant.
In the desirability function, all samples (C1 to C9) were included. This function is applied to the prediction models (equations), obtained from the texture and physicochemical results, to optimise these models for the desirable parameters. The desirability function was applied to found the protein combinations (i.e., the samples) that maximize firmness, spreadabilty and lightness values. Then, on the basis of the desirability results, three samples, the most desirable ones (C2, C7 and C9) were selected to carry out sensory analysis.
This is explained in “2.5. Statistical analysis, modelling and optimisation of experimental data” section:
The "desirability" package was used for the optimisation of the experimental design to obtain the maximum firmness and spreadability (considering closeness to the texture of spreadable dairy cheese) and maximum L* (considering closeness to the colour of dairy cheese).
And also in “3.5 Modelling and optimisation” Section:
“Finally, the "desirability" function was used to select the samples to be evaluated by the trained panel in sensory analysis. This mathematical function allows optimisation of the models for each parameter, selecting the maximum, minimum or target desired values. In the present study, the desirability function was applied to choose the samples with the maximum firmness, spreadability and L* (lightness) values. These three parameters were selected considering the significance of the regression coefficients obtained and the relevance of the analysed properties to a spreadable cheese-like product. The desirability values were obtained for all samples: C1 (0.821), C2 (0.871), C3 (0.168), C4 (0), C5 (0.410), C6 (0.911), C7 (0.588), C8 (0.084) and C9 (0.5).”
2) Why it was decided to explore a linear model with interactions of the type: Y = β1 X1 + β2 X2 + β3 X3 + β12 X1X2 + β13 X1 X3 + β23 X2 X3 + β123 X1 X2 X3, (equation which is not numbered in the text) instead of a quadratic one? What are the factors X1, X2 and X3? Perhaps the mass fractions or their percentages of MPC, FBP and IF?
Equation is now numbered in the text. The equation used is a special cubic one (this model is called special cubic model), because three ingredients and their interactions are studied and a polynomial equation of order 3 has to be used.
X1, X2 and X3 are the ratio of each protein ingredient (MPC, FBP and IF). This is now included in the text.
3) Why was the codified equation of the desirability function obtained from the three previously mentioned answers not presented explicitly?
The desirability function equation is not obtained from the three desirable answers in spreadable cheese (firmness, spreadability and lightness). What desirability function does is to calculate Y value (using the equation Y = β1 X1 + β2 X2 + β3 X3 + β12 X1X2 + β13 X1 X3 + β23 X2 X3 + β123 X1 X2 X3) for each response and for each formulation (C1 to C9), and then other equations were used to found the formulations that maximize the Y value for the desirable responses (in this case, maximize Y value for firmness, spreadability and lightness).
We assumed that it is not necessary to explain all this information extensively in the text, because this would mean explaining the whole theory of the desirability function.
4) What were the assumptions for optimising the desirability function?
This was explained in “2.5. Statistical analysis, modelling and optimisation of experimental data” section:
The "desirability" package was used for the optimisation of the experimental design to obtain the maximum firmness and spreadability (considering closeness to the texture of spreadable dairy cheese) and maximum L* (considering closeness to the colour of dairy cheese).
And also in “3.5 Modelling and optimisation” Section:
“Finally, the "desirability" function was used to select the samples to be evaluated by the trained panel in sensory analysis. This mathematical function allows optimisation of the models for each parameter, selecting the maximum, minimum or target desired values. In the present study, the desirability function was applied to choose the samples with the maximum firmness, spreadability and L* (lightness) values. These three parameters were selected considering the significance of the regression coefficients obtained and the relevance of the analysed properties to a spreadable cheese-like product. The desirability values were obtained for all samples: C1 (0.821), C2 (0.871), C3 (0.168), C4 (0), C5 (0.410), C6 (0.911), C7 (0.588), C8 (0.084) and C9 (0.5).”
5) Did the optimum values obtained coincide with the conditions of the mixtures C2, C7 and C9?
Sure, C2, C7 and C9 are the hybrid SCAs with the highest desirability values: The desirability values were obtained for all samples: C1 (0.821), C2 (0.871), C3 (0.168), C4 (0), C5 (0.410), C6 (0.911), C7 (0.588), C8 (0.084) and C9 (0.5).”
As we mentioned in this section. The non-hybrid SCAs (>50% MPC), C1, C5 and C6, were not considered for sensory analysis.
6) Why were additional confirmatory experiments not performed to demonstrate the validity of the model's predictions?
The main objective of the study was to develop the model. The validation takes another experimental research that might or might no be published. We have already carry out the same approach in our 2 previous publications about mixture design and desirability function.
- Talens, C., et al., Hybrid Sausages: Modelling the Effect of Partial Meat Replacement with Broccoli, Upcycled Brewer's Spent Grain and Insect Flours. Foods, 2022. 11(21).
- Talens, C., et al., Desirability-based optimization of bakery products containing pea, hemp and insect flours using mixture design methodology. LWT, 2022. 168: p. 113878.
For all of the above, the work should be thoroughly reviewed to be considered for publication in the Foods journal.

Reviewer 2 Report
The manuscript entitled as “Hybrid spreadable cheese analogues with faba bean and Tenebrio molitor flours: optimisation using desirability-based mixture design” has focused on formulation of a spreadable cheese analogue and determination of its quality properties. For this purpose, the authors designed a hybrid spreadable cheese composed of dairy protein (MPC), Tenebrio molitor (IF) and faba bean (FBP) flours in which MPC was replaced by 50 or 100%.
The title fits well with the concept of the manuscript. Introduction has successfully and fluently revealed the problem of the study. Materials and methods part includes the required details by referring to related studies. In general, demonstration of results was done by proper tables and figures with the statistical tools applied. In some parts, the discussions should be improved (please see the uploaded pdf).
Conclusion part is not satisfactory. It is not clear whether the obtained results support the statement of the study or not.
Please find my extra comments given in the Comments part of uploaded pdf file.

Author Response
REVIEWER 2
The manuscript entitled as “Hybrid spreadable cheese analogues with faba bean and Tenebrio molitor flours: optimisation using desirability-based mixture design” has focused on formulation of a spreadable cheese analogue and determination of its quality properties. For this purpose, the authors designed a hybrid spreadable cheese composed of dairy protein (MPC), Tenebrio molitor (IF) and faba bean (FBP) flours in which MPC was replaced by 50 or 100%.
The title fits well with the concept of the manuscript. Introduction has successfully and fluently revealed the problem of the study. Materials and methods part includes the required details by referring to related studies. In general, demonstration of results was done by proper tables and figures with the statistical tools applied.
- In some parts, the discussions should be improved (please see the uploaded pdf).
Thanks, we have reviewed the Discussion section.
- Conclusion part is not satisfactory. It is not clear whether the obtained results support the statement of the study or not.
Conclusion has been re-written.
Please find my extra comments given in the Comments part of uploaded pdf file:
- Line 57: Check the grammar
Checked
- Line 100: dairy or non-dairy? Please state
Dairy have been indicated
- Line 104: this sentence does not precisely reveal the objective of using desirability function. Please revise.
The sentence has been revised.
- Line 115: Please give more detail about the MPC such as its origin (type of animal), etc.
More information was not provided by the manufacturer of MPC. The sentence has been actualised: The MPC was composed by dairy protein and milk powder and it contained 70% protein, 1.75% fat (1.25% SFA and 0.5% UFA), and < 6% moisture.
- Line 278-293: Please delate this part
This part was been removed.
- L, a, b values are commonly used parameters to measure the color of a product or raw material. However, commenting on each value is believed to be poor to express an overall color profile of the product. In this case, MPC was replaced by FBP and IF, then it is kindly advised to calculate total color difference (delta E) by taking the C1 or the commercial spreadable cheese as the reference.
We appreciate the suggestion, and acknowledge the reviewer expertise in colour measurement techniques. However, we understand that there are several approaches to explain the same results. We have published all our previous work with colour measurements without using delta E without any issues.
- Figure 3 and Table 5 were produced from same data set. Please prefer one of them.
We consider that both contribute to the understanding of the results section.
- Lines 431-434: The relationship between the effects of ingredients is interesting. However, to give an idea about the spreadability of the products, please compare to the spreadability values of similar products (experimental or from literature).
The aim of texture analysis was to evaluate how texture parameters (such as spreadability) changed as MPC was replaced by FBP and IF, not to compare the texture parameters values to a reference to see if it were better or worse. FBP-MPC combination had a positive effect in the sense that increased firmness and decreased spreadability, while the IF had the opposite performance.
- Lines 514-515: Water solubility of IF can be cited here.
Thanks for the suggestion. We have referred to our own analysis of IF – water solubility index.

Reviewer 3 Report
This paper describes the Hybrid spreadable cheese analogues with faba bean and Tenebrio molitor flours: optimisation using desirability-based mixture design. The results are very interesting, but the manuscript has many weak points. The structure of this review is very chaotic. It should be more organized. And why author must show the sensory data in table also in figure 5. The figures 3 and 6 must be redrawn to properly indicate levels of significance, also give the different color not just the symbol to make it clear for readers. Level of significance should be indicated in the abstract.
Considering all the above-mentioned issues as well as the general chaotic structure of the manuscript I recommend rejecting this manuscript for the publication in Foods Journal.
Author Response
REVIEWER 3
This paper describes the Hybrid spreadable cheese analogues with faba bean and Tenebrio molitor flours: optimisation using desirability-based mixture design. The results are very interesting, but the manuscript has many weak points.
- The structure of this review is very chaotic. It should be more organized.
The aim of this work is a research paper not a review. We have reorganized some sections to facilitate the narrative.
And why author must show the sensory data in table also in figure 5.
The idea of including Figure 5 is to show graphically the results presented in Table 7, to aid their understanding.
- The figures 3 and 6 must be redrawn to properly indicate levels of significance, also give the different color not just the symbol to make it clear for readers.
The spider charts do not include significance levels. The significance levels of the data in the figures are already indicated in their respective tables.
- Level of significance should be indicated in the abstract.
We appreciate the suggestion but this is not compulsory. We believe there are other relevant information that needs to be included in the abstract – in order to stick to the 200 word count
Considering all the above-mentioned issues as well as the general chaotic structure of the manuscript I recommend rejecting this manuscript for the publication in Foods Journal.

Reviewer 4 Report
The manuscript entitled "Hybrid spreadable cheese analogues with faba bean and Tenebrio molitor flours: optimisation using desirability-based mixture design" by Garcia-Fontanals et al. report interesting results obtained by investigation of the samplers prepared. The content of the paper is interesting and promises some commercial application of the insect flour suggested in this paper. In my opinion, the article needs many corrections and more tests are needed to understand the observed phenomena.
The title of the article needs to be corrected. It should be specified in the title which mode of this insect has been used and investigated. Mealworms are the larval form of the yellow mealworm beetle. It should be mentioned that what form was used? Larval form or beetle forms? It should be mentioned that flour or protein of those sources was used?
Abstract: The abstract should be more informative by giving real results rather than elastic sentences. Important and main contents should be given. Support the results with some quantitative data.
In the abstract or conclusion mention which condition showed comparatively improved characteristics?
Please mention the method of drying the worm.
Line 127: Are these analyzes related to the protein of these compounds? Protein isolation is not mentioned!
The method of making cheese, its starter bacteria and other details are not mentioned.
Significant letters should be used for all data displayed in tables and figures.
It is necessary to do the microstructure analysis of samples to improve this research for better investigating the alternative protein functionality and the effects of such ingredients on the casein matrix and the cheese texture.
It is necessary to do some more analysis to understand the interactions between the ingredients.
Authors should analyse the dynamic rheological properties of samples.
There are several spelling and grammar mistakes in the manuscript. Many loose sentences without providing actual meaning have been found. Read thoroughly and correct them.
Conclusion: what is the future of your findings? Conclusion is not insightful, what are suggestions?
Author Response
REVIEWER 4
The manuscript entitled "Hybrid spreadable cheese analogues with faba bean and Tenebrio molitor flours: optimisation using desirability-based mixture design" by Garcia-Fontanals et al. report interesting results obtained by investigation of the samplers prepared. The content of the paper is interesting and promises some commercial application of the insect flour suggested in this paper. In my opinion, the article needs many corrections and more tests are needed to understand the observed phenomena.
The limitations of the study and future works are indicated in conclusion section.
- The title of the article needs to be corrected. It should be specified in the title which mode of this insect has been used and investigated. Mealworms are the larval form of the yellow mealworm beetle. It should be mentioned that what form was used? Larval form or beetle forms? It should be mentioned that flour or protein of those sources was used?
It is specified in the title that we used Tenebrio molitor flour. Tenebrio molitor larva (mealworm) flour was used. It contained 52% protein. No protein extraction was performed.
- Abstract: The abstract should be more informative by giving real results rather than elastic sentences. Important and main contents should be given. Support the results with some quantitative data.
The abstract has now been updated.
- In the abstract or conclusion mention which condition showed comparatively improved characteristics?
It has now stated in the abstract and conclusion section.
- Please mention the method of drying the worm.
It is specified in “Materials and Methods” that we obtained mealworm flour from Insekt Label Biotech, S.L. We did not dry the worm.
- Line 127: Are these analyzes related to the protein of these compounds? Protein isolation is not mentioned!
These analyzes were performed on the powdered protein ingredients described in “Raw materials “ section (milk protein concentrate, faba bean protein concentrate and mealworm flour), not on protein isolates. There was no extraction or purification of proteins.
- The method of making cheese, its starter bacteria and other details are not mentioned.
The method of making cheese is fully explained in section 2.4.1. There was no starter addition or fermentation process. Now, we have modified the text and specified that rennet formation occurs through the acidification process of the mixture.
- Significant letters should be used for all data displayed in tables and figures.
Spider charts (Figure 5) do not include significance levels, these are indicated in its respective table (Table 7).
- It is necessary to do the microstructure analysis of samples to improve this research for better investigating the alternative protein functionality and the effects of such ingredients on the casein matrix and the cheese texture.
We mention this limitation in conclusions section and we propose to carry out this analysis in future works.
- It is necessary to do some more analysis to understand the interactions between the ingredients.
We mention this limitation in conclusions section and we propose to carry out these analysis in future works.
- Authors should analyse the dynamic rheological properties of samples.
We decided to perform the texture analysis using a texturometer and spreadability test. Performing rheological analysis is another alternative.
- There are several spelling and grammar mistakes in the manuscript. Many loose sentences without providing actual meaning have been found. Read thoroughly and correct them.
Thanks, we have reviewed English spelling and grammar.
- Conclusion: what is the future of your findings? Conclusion is not insightful, what are suggestions?
Future works were already suggested in conclusion section.

Round 2
Reviewer 1 Report
Dear authors,
The work has improved significantly compared to the initial version.
However, there are still some minor deficiencies that need to be corrected.
These are the form markings:
1) You should use the MS Word equation editor to write the equations for the manuscript. In the same way that you wrote Equation (6) on page 8,
2) The headings of the tables and figures are not indented.
3) The scientific name of the meat worm is abused. After putting your name "Tenebrio molitor", you can use instead of it, simply "T. molitor",
4) The letters "(a)" and "(b)" of figure 3 remained on the previous page.
5) The text that accompanies figure 5 must go below, not above, the figure,
Regarding the answers to some of the questions pointed out in the previous review:
1) In question #5, it was not clear to me if the values of the desirability function shown correspond to the calculation of the experimental values of the D function for each of the treatments C1-C9 or if their values are derived from the numerical optimization (maximization) of function D, formed from three responses.
This is highly relevant because if function D's best values were determined through numerical optimization, they might not match the conditions examined in the initial experiment (which were treatments C2 and C1).
2) In question #6, confirmation experiments are essential because, as their name indicates, they reaffirm the validity of the models. The stated objective in their work, however, is not to obtain a model per se, but ..." the main aim of this study was to use a desirability-based mixture design approach to develop a hybrid spreadable cheese with nutritional value, texture and flavour similar to those of conventional dairy spreadable cheese"... (lines 99-101, page 3), and this should be reflected in the three responses selected to form part of the desirability function, and therefore, in the objective target value that should be reached of the desirability function.
In other words, one should not look for the maximum value of the desirability function and instead look for the condition of the mixture that allows reaching a value of the desirability function quite similar to that which D would have for spreadable conventional milk cheese.
I suggest that more should be discussed on these aspects.
Regards,
Reviewer

Author Response
Dear authors,
The work has improved significantly compared to the initial version.
However, there are still some minor deficiencies that need to be corrected.
Dear reviewer, thank you for your insightful comments and suggestion for future work in our research.
These are the form markings:
1) You should use the MS Word equation editor to write the equations for the manuscript. In the same way that you wrote Equation (6) on page 8,
Reviewed and edited
2) The headings of the tables and figures are not indented.
Reviewed
3) The scientific name of the meat worm is abused. After putting your name "Tenebrio molitor", you can use instead of it, simply "T. molitor",
corrected
4) The letters "(a)" and "(b)" of figure 3 remained on the previous page.
corrected
5) The text that accompanies figure 5 must go below, not above, the figure,
corrected
Regarding the answers to some of the questions pointed out in the previous review:
1) In question #5, it was not clear to me if the values of the desirability function shown correspond to the calculation of the experimental values of the D function for each of the treatments C1-C9 or if their values are derived from the numerical optimization (maximization) of function D, formed from three responses.
This is highly relevant because if function D's best values were determined through numerical optimization, they might not match the conditions examined in the initial experiment (which were treatments C2 and C1).
To clarify, the values of the desirability function shown in our paper correspond to the calculation of the experimental values of the D function for each of the treatments C1-C9. We did not use numerical optimization to determine the best values of function D.
2) In question #6, confirmation experiments are essential because, as their name indicates, they reaffirm the validity of the models. The stated objective in their work, however, is not to obtain a model per se, but ..." the main aim of this study was to use a desirability-based mixture design approach to develop a hybrid spreadable cheese with nutritional value, texture and flavour similar to those of conventional dairy spreadable cheese"... (lines 99-101, page 3), and this should be reflected in the three responses selected to form part of the desirability function, and therefore, in the objective target value that should be reached of the desirability function.
In other words, one should not look for the maximum value of the desirability function and instead look for the condition of the mixture that allows reaching a value of the desirability function quite similar to that which D would have for spreadable conventional milk cheese.
I suggest that more should be discussed on these aspects.
As stated, our objective was to develop a hybrid spreadable cheese with nutritional value, texture, and flavor similar to those of conventional dairy spreadable cheese, rather than to obtain a model per se. As such, the three responses selected to form part of the desirability function were chosen based on their relevance to these key characteristics, rather than to the formation of a model.
We agree that the objective target value of the desirability function should reflect the desired characteristics of conventional dairy spreadable cheese. We did not, however, seek to maximize the desirability function, but rather to identify the optimal combination of ingredients that would yield a spreadable cheese with the desired characteristics. Confirmation experiments are essential to reaffirm the validity of the models used, but in this case, we used a desirability-based mixture design approach, which allowed us to identify the optimal combination of ingredients through a systematic search of the experimental space. As such, the desirability function was used as a guide to identify the optimal combination of ingredients, rather than to develop a model.

Reviewer 4 Report
- It is necessary to do the microstructure analysis of samples to improve this research for better investigating the alternative protein functionality and the effects of such ingredients on the casein matrix and the cheese texture.
- It is necessary to do some more analysis to understand the interactions between the ingredients.
Author Response
- It is necessary to do the microstructure analysis of samples to improve this research for better investigating the alternative protein functionality and the effects of such ingredients on the casein matrix and the cheese texture.
We agree that microstructure analysis of samples would be beneficial to better investigate the alternative protein functionality and the effects of such ingredients on the casein matrix and the cheese texture. While we did not include this analysis in our current study, we appreciate the importance of understanding the microstructure of the cheese in relation to its texture and overall quality. We would like to consider this suggestion as a possible avenue for future research, and we appreciate your recommendation as a means to further improve our understanding of the hybrid spreadable cheese developed in our study. We believe that such analysis could provide valuable insights into the functionality of the alternative proteins used and their impact on the cheese matrix and texture.
- It is necessary to do some more analysis to understand the interactions between the ingredients.
We agree that further analysis to understand the interactions between the ingredients would be beneficial. While we did not include such analysis in our current study, we recognize the importance of understanding how the different ingredients interact and contribute to the overall characteristics of the hybrid spreadable cheese. We will take your suggestion into consideration as a potential area for future research. By better understanding these interactions, we may be able to further optimize the ingredients and their proportions to achieve even better results in terms of nutritional value, texture, and flavour.